# Neuropathology, Neuroimaging, and Fluid Biomarkers in Alzheimer’s Disease

**DOI:** 10.3390/diagnostics14070704

**Published:** 2024-03-27

**Authors:** Helena Colvee-Martin, Juan Rayo Parra, Gabriel Antonio Gonzalez, Warren Barker, Ranjan Duara

**Affiliations:** 1Wien Center for Alzheimer’s Disease & Memory Disorders, Mount Sinai Medical Center, Miami Beach, FL 33140, USA; helenacolvee@gmail.com (H.C.-M.); warren.barker@msmc.com (W.B.); 2Human & Molecular Genetics, Florida International University, Miami, FL 33199, USA; jrayo006@fiu.edu (J.R.P.); ggonz288@fiu.edu (G.A.G.)

**Keywords:** Alzheimer’s, PET scan, MRI, plasma biomarkers, amyloid beta protein, hyper-phosphorylated tau, neuropathology

## Abstract

An improved understanding of the pathobiology of Alzheimer’s disease (AD) should lead ultimately to an earlier and more accurate diagnosis of AD, providing the opportunity to intervene earlier in the disease process and to improve outcomes. The known hallmarks of Alzheimer’s disease include amyloid-β plaques and neurofibrillary tau tangles. It is now clear that an imbalance between production and clearance of the amyloid beta protein and related Aβ peptides, especially Aβ42, is a very early, initiating factor in Alzheimer’s disease (AD) pathogenesis, leading to aggregates of hyperphosphorylation and misfolded tau protein, inflammation, and neurodegeneration. In this article, we review how the AD diagnostic process has been transformed in recent decades by our ability to measure these various elements of the pathological cascade through the use of imaging and fluid biomarkers. The more recently developed plasma biomarkers, especially phosphorylated-tau217 (p-tau217), have utility for screening and diagnosis of the earliest stages of AD. These biomarkers can also be used to measure target engagement by disease-modifying therapies and the response to treatment.

## 1. Introduction

Alzheimer’s disease (AD) is a progressive neurodegenerative disease that typically presents with memory deficits for recent events, although other cognitive deficits and clinical presentations have been recognized, including progressive language deficits, neuropsychiatric features (such as depression, apathy, paranoid delusions, and irritability) as well as motor changes such as bradykinesia [1]. The most prominent risk factors for AD are age (typically 60+ years), family history of the disease [2], female sex [3], cardiovascular risk factors, and diabetes mellitus [4]. Although the frequency and severity of cardiovascular disease have decreased overall since the 1960s with advances in modern medicine [5], the resulting aging of the population, worldwide, constitutes a growing public health concern with respect to the burden of AD [6]. Thus, it remains important to reduce the prevalence and severity of the disease by developing methods for early diagnosis and early intervention through an improved understanding of the pathobiology of AD.

In 1984, (McKhann et al.) [7], the first internationally accepted clinical criteria for Alzheimer’s disease, known as the NINCDS-ADRDA criteria for Alzheimer’s disease, were published. These criteria recognized the typical and progressive clinical features of the disease, although the absence of any available biomarkers of the disease, at the time, made it necessary to rely solely on ruling out other conditions that mimicked the clinical features of AD. Clinical confirmation of the diagnosis by neuropathological diagnosis at autopsy was about 70 to 90% accurate [8]. In 2011, the National Institute on Aging and Alzheimer’s Association created a “research framework”, also known as the AT(N) framework, for the preclinical, mild cognitive impairment (MCI), and dementia stages of AD [9,10]. The AT(N) research framework focused on biomarkers in living persons, grouped into those representing β amyloid (Aβ) or “A”, pathologic tau or “T”, and neurodegeneration or “(N)”, which is defined by the underlying pathologic processes in the pathology of AD, that define the disease at postmortem examination and can be assessed in vivo by biomarkers. Parentheses are used for “(N)” because neurodegeneration is not specific to AD but is an essential aspect of the pathological aspect of the disease.

The first detectable pathology of AD in the brain, which is amyloid β (Aβ) protein in extracellular plaques, occurs decades before the onset of clinical symptoms [11]. This is followed by the hyperphosphorylation of tau resulting in neurofibrillary tangles [12], and neuronal degeneration, which can be detected on structural MRI brain scans [13]. Biomarkers for amyloid and tau, neurodegeneration, and inflammatory changes in cerebrospinal fluid (CSF) have provided an invaluable foundation of knowledge for understanding the earliest molecular changes that occur in Alzheimer’s disease, as well as a reliable method for establishing the diagnosis of AD [14]. However, new ultra-sensitive assays for plasma have enabled the accurate measurement of A, T, and (N) in blood samples replacing the need for lumbar puncture and reducing the use of neuroimaging in research and the clinic [15].

In 2018, the NIA-AA research framework for the biological definition and diagnosis of AD was updated to recognize the methodological and conceptual progress that had been made as a result of validating the ATN biomarker framework [16]. The methodological validation of the AT(N) framework demonstrated that it represents the core components of AD pathology, in a continuum from the earliest stages of the disease to the dementia stage. The conceptual progress was to identify the AT(N) framework as not only reflecting the core components of AD pathology but as being independent of the clinical stage, such that the underlying pathology is not necessarily associated with any clinical consequences. Eight different AT(N) profiles were studied according to the combination of biomarker profiles; four patterns of progression emerged, including the classical AT(N) sequence, which was the most frequent [17]. In 2023, a further revision of the NIA-AA criteria was drafted to reflect both the research and the potential clinical utility of the ATN framework. This recent update will reflect the development of disease-modifying treatments and the growing utility of plasma biomarkers for the clinical diagnosis of AD and for monitoring the response to treatment [18].

In the following sections, we examine the foundations for the diagnosis of Alzheimer’s disease in life. The journey to our current state of knowledge began with developing a detailed understanding of the pathology of AD. This was followed by the development of imaging tools to identify structural, metabolic and biochemical changes in the brain in AD. Finally, the ability to detect various proteins representing different components of the pathology of AD has brought us to the brink of revolutionizing the diagnosis of AD. Initially, these fluid biomarkers assessments were restricted to analyzing various proteins in the cerebrospinal fluid. However, increasing accuracy in the detection of these proteins in the plasma has brought us to the brink of screening for AD pathology, and subsequently diagnosing and staging underlying AD pathology, using blood biomarkers alone. The ATN profile in an individual patient may also provide a measure of the prognosis in individual patients, which will be enhanced by structural and metabolic imaging to identify additional pathological entities, such as vascular disease, space-occupying and inflammatory lesions, and hydrocephalus.

In this review, we have avoided an exhaustive account of the inter-relatedness within, and between, neuropathological, imaging, and biomarker features, as well as the use of these features for enabling a differential diagnosis of different causes of cognitive and functional impairment. Instead, the overall goal of this review is to outline and describe the underlying pathological features of AD and the potential of available imaging and fluid biomarkers for diagnosing AD at an early, preclinical stage, and for tracking the progression of the disease and its response to disease-modifying treatments.

## 2. Neuropathology of Alzheimer’s Disease

The two neuropathological changes in the brain required for a diagnosis of Alzheimer’s disease are amyloid plaques and neurofibrillary tangles [19]. Amyloid and tau are naturally occurring protein aggregates that become pathogenic when they become structurally abnormal, a process known as misfolding [20]. Deposits of amyloid plaques and accumulation of neurofibrillary tangles occur early in the disease process—years or decades before symptoms occur—and lead to neurodegeneration and cognitive decline [11]. While AD pathology is the most frequent finding in people with dementia who come to autopsy, there is a high rate of concomitant pathology including Lewy body pathology, amyloid angiopathy, cerebrovascular disease, and hippocampal sclerosis [21]. These additional pathologies accelerate the rate of cognitive decline in people with MCI [22].

### 2.1. Senile Plaques (SP)

SPs are extracellular nonvascular deposits of beta-amyloid protein (Aβ) that initially develop in areas of the brain such as the temporo-basal cortex, anterior cingulate regions, parietal, and frontal neocortex, as well as through the amygdala, basal ganglia, and hippocampus in the later stages of the disease [23]. Amyloid-ß protein is produced in the brain by astrocytes and neurons. It is a 38–42 amino acid long peptide, derived from a transmembrane protein (Amyloid-β precursor protein or APP) [24]. The altered cleavage of APP by enzymes, such as β-secretases (BACE1) and γ-secretases, results in insoluble (Aβ) fibrils. Consequently, amyloid polymerizes into insoluble protein fibrils that will aggregate and form plaques (Figure 1, [25]). There are two types of beta-amyloid polymers with an essential role in this process: Aβ40 and Aβ42. The first one, Aβ40, is less neurotoxic than the latter, which is the major component of the amyloid plaques and is the most fibrillogenic. Aβ40/Aβ42 aggregation alters homeostasis by blocking specific ion channels, which leads to increased oxidative stress and neuronal cell death [23]. These cellular pathways are potential therapeutic targets.

### 2.2. Neurofibrillary Tangles (NFTs)

The neuropathological diagnosis of Alzheimer’s disease requires both amyloid plaques and neurofibrillary tangles (NFTs). These tangles are abnormal filaments composed of misfolded hyperphosphorylated tau protein. The microtubule-associated protein tau (MAPT) is usually present in the cytoplasm of neuronal axons and has the function of regulating the stabilization of microtubules, which form the axonal cytoskeleton and serve in the intracellular transport of organelles and axon outgrowth [26,27].

Post-translational modifications of tau protein, such as phosphorylation, glycosylation, and acetylation are common in tau proteins and can affect the rate of Alzheimer’s progression, with phosphorylation being the predominant modification. These modifications cause the tau proteins to become hyperphosphorylated and aggregate in dendrites and cell bodies. The progression and evolution of tau proteins are reflected in their different post-translational stages. The hyperphosphorylation of tau protein results in the formation of paired helical filaments (PHF), which have twisted or helical shapes. These PHFs destabilize microtubules and induce the formation of NFTs, which lead ultimately to neuronal dysfunction and death. Tau can occur in six isoforms caused by alternative splicing on different exons such as 2, 3, and 10 [27]. These isoforms receive the denomination 3-Repeat (3R) or 4-Repeat (4R), depending on the number of microtubule-binding domains present. Alzheimer’s disease presents a mixture of both 3R and 4R tau proteins [27].

### 2.3. Vascular Changes in AD

Epidemiological research studies have shown that cardiovascular disease is an important risk factor and contributes to the development of AD pathology [28]. Cerebral vasculature undergoes different pathological changes that influence processes involved in the development and progression of the disease. Risk factors characteristic of both AD and CVD are hypertension, obesity, and diabetes. Vascular dementia (VaD) accounts for approximately 5–10% of patients with dementia, while AD has a percentage of 60–80% [21]. However, most people with dementia at autopsy have a mixed pathology.

The damage to cerebral vasculature can cause malfunction of the neurovascular unit and disrupted cerebral blood flow [29]. The neurovascular unit is a region in the brain containing endothelial cells that function as the blood–brain barrier (BBB), which regulates the brain’s homeostasis. BBB disruption leads to neurodegeneration and reduces the flow of amyloid through the brain, causing amyloid deposition in the walls of blood vessels and capillaries, thereby impairing their function. Early breakdown of the blood–brain barrier has been linked to disrupted microcirculation in the white matter, which in turn results in chronic cerebral hypoperfusion [29]. Hypertension reduces blood flow due to vasoconstriction of the affected vessels and is associated with increased amyloid deposition and the formation of neurofibrillary tangles [30].

### 2.4. Cerebral Amyloid Angiopathy (CAA)

CAA is a hallmark feature of AD that results in alterations of vascular clearance mechanisms. Aβ peptides can deposit in the brain parenchyma as amyloid plaques or in cerebral blood vessels. Research demonstrates that 85–95% of Alzheimer’s disease cases have some degree of cerebral amyloid angiopathy [19]. In CAA, the amyloid deposits are mostly composed of Aβ40, in contrast with the parenchymal deposits, which are rich in Aβ42. CAA affects different blood vessels such as the small arteries, arterioles, and capillaries in the gray matter of the brain [19]. Two types of CAA have been differentiated. Type 1 is more strongly associated with the APOE e4 allele and affects arteries, arterioles, and capillaries. In contrast, type 2 has been associated with the APOE e2 allele and affects both arteries and arterioles but not capillaries [19]. Severe cases of CAA can produce brain hemorrhages, ischemic lesions, and impaired blood flow.

### 2.5. Inflammation, Astrogliosis, and Microglial Activation

Microglial cells, which are resident phagocytes of the CNS, function by monitoring different brain areas for exposure to pathogens and areas of injury, including damaged and dying cells. They contribute to neuronal plasticity and the protection of synapses as well [31]. Once these cells become activated, they migrate to the site of the lesion, where they provide an innate immune response. Microglial activation appears at the prodromal and preclinical stages of Alzheimer’s disease and seems to play a protective role in these early stages. Aβ plaques and NFTs, which are highly toxic, lead to synaptic damage and increased oxidative stress in the brain [24]. In AD, microglial cells deliver part of the inflammatory response by secreting proinflammatory cytokines and binding via cell-surface receptors, Aβ fibrils, and NFT fibrils. Subsequently, microglia engulf the Aβ and NFT fibrils via phagocytosis [31] and eliminate them by endolysosomal degradation.

Inefficient clearance of these misfolded proteins is a major pathogenic pathway that accelerates disease progression. An increased level of cytokines in the lesion area can downregulate the expression of phagocytic receptors involved in the clearance of Aβ. This phenomenon is associated with a rare variant of the TREM2 gene (triggering receptor expressing on myeloid cells) which contributes to the risk for AD almost to the same extent as the APOE e4 allele [32]. The strong correlation between microglial activation and Alzheimer’s disease pathogenesis makes these cells a major therapeutic target.

On the other hand, reactive astrocytes accumulate around senile plaques and neurofibrillary tangles almost to the same extent as microglial cells, and they are often found in post-mortem brain tissue of humans suffering from AD [33]. Astrogliosis has been shown to increase the levels of glial fibrillary acidic protein or GFAP, which is associated with the severity of cognitive and functional impairment [34]. Similar to microglial cells, when astrocytes are exposed to amyloid, they also release cytokines, interleukins, and other cytotoxic molecules that regulate and exacerbate the neuroinflammatory response. Cytokines modulate both microglial activation and astrogliosis, thereby contributing to the process of neuroinflammation, as pro- and anti-inflammatory processes.

### 2.6. Glucose Hypometabolism

Glucose is the brain’s main source of energy. The brain utilizes 20% of the body’s total oxygen consumption and 25% of the body’s total glucose [35] both at rest and in the awake state. Evidence suggests that abnormal glucose metabolism is present in Alzheimer’s disease, and its decline leads to impaired brain function [36]. Glucose metabolism in the brain entails two main processes: intracellular oxidative metabolism and glucose transportation [35]. Transportation of glucose largely depends on the function of astrocytes and the composition of the blood–brain barrier, as well as on the numerous glucose transporters distributed throughout this organ. The glucose transporters more directly involved with glucose hypometabolism in Alzheimer’s disease are GLUT-1 and GLUT-3, which modulate glucose transportation and contribute to AD pathogenesis [36]. Impairment of glucose transport and glucose hypometabolism, as well as insulin resistance and type 2 diabetes mellitus, with resulting mitochondrial dysfunction, are associated with abnormal cognitive function in AD patients. The PET tracer, [F-18] fluorodeoxyglucose (FDG), and FDG-PET studies have demonstrated a correlation between progressive and consistent glucose hypometabolism and symptom severity in AD [37]. The regional cortical hypometabolism characteristic of AD likely reflects the ongoing pathology in medial temporal brain regions such as the entorhinal cortex and hippocampus [37].

### 2.7. Glymphatic System Impairment

The glymphatic system is considered a waste clearance or removal system composed of perivascular channels. These channels are made of astroglia cells and their main function is to eliminate biogenic substances and soluble proteins from the central nervous system [38]. The glymphatic system serves as the replacement for a conventional lymphatic system, which is absent in the central nervous system. There is accumulating evidence that dysfunction in the glymphatic system, with the resulting delayed clearance of waste products, may be the pathogenesis of idiopathic normal pressure hydrocephalus [39]. Multiple human studies have brought to light correlations between the impairment of the glymphatic system with the progression of AD pathology [40]. The degradation of glial cells and neurons, impairment of glymphatic system flow, as well as transport across the blood–brain barrier, are all considered contributing mechanisms to the removal of Aβ plaques [41].

The degradation of glial cells and neurons, glymphatic system flow, as well as transport across the blood–brain barrier, are all considered contributing mechanisms to the removal of Aβ plaques [41]. However, patients suffering from AD present with altered CSF dynamics [42], which leads to the accumulation of amyloid plaques. This may indicate that AD can cause a vicious cycle in which plaque aggregation along blood vessels reduces the efficiency of the glymphatic system, leading to further neuronal death [40].

### 2.8. White Matter Changes in AD

Alzheimer’s disease is known to cause two main progressive pathological changes in the gray matter of the brain: amyloid plaques and NFTs. However, multiple neuroimaging studies have shown micro- and macro-structural changes in white matter that have been linked to the development of AD [43]. There is pathological evidence for the toxic effects of beta-amyloid on white matter in myelin, including the loss of axons and oligodendrocytes, through mechanisms such as reactive astrocytosis, enlargement or dilation of the perivascular space, and the failure of drainage of interstitial fluid secondary to the deposition of beta-amyloid [44]. White matter changes in AD have also been associated with vascular risk factors, vascular insufficiency, as well as Wallerian degeneration of fiber tracts, secondary to neuronal loss in cortical associative areas [45,46,47]. Clinical correlates of white matter changes have been reported to include depression and a decline in motor function, including a slowing of the speed and decline in fine motor coordination, in patients diagnosed with vascular dementia, dementia with Lewy bodies, psychiatric disorders, and other neurodegenerative disorders [48]. These studies suggest that white matter demyelination and degeneration are also deeply involved with the pathogenesis and progression of dementia in Alzheimer’s disease. In a pivotal study, Lee et al. (2016) demonstrated [49] that in autosomal-dominant AD cases, with early age of onset and low likelihood of comorbid vascular disease, there is an increase in white matter hyperintensities (WMH) seen in MRI scans well before the expected symptom onset. The findings suggest the possibility that “WMHs are a core feature of AD, a potential therapeutic target, and a factor that should be integrated into pathogenic models of the disease”. In transgenic animal models of AD, “white matter pathology emerges before the appearance of cortical plaques and tangles” [43,50].

Eloyan et al. (2023) compared the severity of (WMHs) between subjects with amyloid-positive early-onset Alzheimer’s disease (A+EOAD) (*n* = 205) and individuals with normal cognition (CN) (*n* = 89) and individuals who were cognitively impaired but amyloid-negative EOAD (A-EOAD) in the Longitudinal Early-Onset Alzheimer’s Disease Study (LEADS) [51]. A+EOAD showed greater severity of WMHs compared with CN and A-EOAD participants across all regions, with no significant differences between CN and A-EOAD participants. Greater WMHs were associated with worse cognition. Tau burden was positively associated with WMH burden in the A+EOAD group. Overall, greater WMHs were associated with worse cognition and a higher tau burden in EOAD. Oligodendrocytes are the cells responsible for myelin production and maintenance [52]; myelin loss resulting from the impairment of oligodendrocytes to repair myelin damage may represent a main feature of AD [43]. In transgenic mice carrying the Presenilin-1 mutations, oligodendrocytes were sensitized to glutamate and amyloid toxicities, which exacerbated white matter damage and memory impairment in these mice [52]. The study by Eloyan et al. (2023) suggests that the abnormalities of oligodendrocytes occur in the presence of a PS1 mutation. In conclusion, white matter disease is highly prevalent in AD patients, and both degenerative disease and vascular disease are associated with WMHs in AD, although the earliest changes in the white matter on MRI scans in AD patients appear to be related to the onset of neurodegenerative disease (Figure 2).

## 3. Neuroimaging and Fluid Biomarkers in Aging and Dementia

### 3.1. Structural MRI

MRI-based markers of brain atrophy are reliable indicators of the neurodegeneration that typically takes place in Alzheimer’s disease (AD) and its progression. Neuropathological studies show that the pathological features of AD may be present for years before clinical symptoms are evident and that the sites and severity of atrophy strongly correlate with the cognitive decline observed cross-sectionally and longitudinally [13]. These structural changes align accurately with the Braak stages of neurofibrillary tangle deposition and reflect downstream neuropsychological deficits. The initial sites of tau deposition and MRI-visible atrophy often occur in the hippocampal pathway, aligning with initial memory deficiencies. The subsequent atrophy in temporal, parietal, and frontal regions correlates with cognitive and behavioral impairments. Alterations in the brain’s structure, such as hippocampal atrophy and neocortical atrophy, especially in the parietal/precuneus regions, are early markers of neurodegeneration in AD that correlate with the presence and severity of cognitive impairment [53,54].

Heterogeneity in the form of distinct subtypes of brain atrophy in AD closely matches findings reported in autopsy studies. These subtypes of atrophy (typical, limbic-predominant, and hippocampal-sparing) have been identified by Murray et al. [55] in pathological studies. Volumetric MRI reflects the severity of underlying neurodegeneration, can be quantified easily, and is more amenable to identifying such subtypes of AD [56,57]. Volumetric measures from MRI scans can be used to classify these neurodegenerative subtypes along a continuum between limbic-predominant and hippocampal-sparing [57]. In these studies, the purpose of using amyloid imaging and ensuring amyloid positivity, so as to identify subtypes of AD by discrete patterns of regional atrophy on volumetric measures, was precisely to ensure that all the patients in the study did have Alzheimer’s disease, rather than other conditions mimicking AD.

It has been shown that these subtypes of atrophy in AD predict which patients are likely to have earlier onset and shorter disease duration (i.e., rapid progression), as well as the likelihood of being APOE ε4-positive [57]. Using volumetric MRI, Persson et al. [58] demonstrated four different subtypes of atrophy in 123 patients with mild Alzheimer’s disease dementia (AD), including a “typical subtype” in 48%, “limbic-predominant subtype” in 24%, “hippocampal-sparing subtype” in 15%, and a subtype with “minimal atrophy” (previously referred to as no-atrophy AD) in 13%. Although, in this study, no cognitive differences were found and progression rates were similar between the different subtypes, the minimal-atrophy subtype group was found to be less educated and had greater functional impairment (as reflected in the baseline Clinical Dementia Rating sum of boxes scores), but higher levels of Aβ in the cerebrospinal fluid (indicating lower levels of Aβ in the brain). These findings indicated the likelihood of greater vulnerability to the effects of Aβ in subjects with low cognitive reserve. Patterns of cortical atrophy, also measurable on volumetric MRI, correlate with patterns of cognitive impairment, reflecting focal cortical syndromes, such as the frontal-behavioral variant of AD, progressive aphasia, or posterior cortical atrophy syndrome [59].

High-resolution MRI not only correlates with tau pathology but also provides a valuable measure for tracking the stages and progress of AD. At the mild cognitive impairment (MCI) stage, the degree of atrophy in medial temporal structures like the hippocampus has been endorsed as a diagnostic indicator (Figure 3). In this regard, Jack et al. [54] found that a 2-SD reduction in hippocampal volume (corrected for age and sex) was associated with a fourfold increase in the percentage of individuals with MCI converting to dementia within 5 years. Various MRI sequences can capture both macrostructural and microstructural changes in the brain, including dendritic, axonal, and myelin loss and metabolic alterations. Techniques like magnetic resonance spectroscopy, diffusion-weighted imaging, fiber tracking, and magnetization transfer imaging add valuable layers of information. While some of these methods, like arterial spin labeling and functional measures of resting-state networks, show promise as diagnostic markers, they still require rigorous validation [60,61]. Regarding disease progression, the rate of change in structural markers like whole-brain, entorhinal cortex, hippocampus volumes, and ventricular enlargement correlates closely with shifts in cognitive performance. Effective clinical utilization of these atrophy markers requires a comprehensive understanding of their behaviors across different stages of AD and how they interact with other imaging and biological markers. These structural indicators are susceptible to changes across a broad disease severity spectrum, from MCI to moderate dementia stages. However, in the transition from asymptomatic to MCI stages, markers of amyloid pathology often show more pronounced abnormalities than structural markers [62,63].

Magnetic resonance imaging (MRI) scans are widely used to evaluate elderly patients with cognitive complaints. The primary clinical purpose is often to use information derived from MRI scans to identify or exclude the presence of various lesions that could contribute to the cause of cognitive impairment and dementia, including vascular lesions, hematomas, neoplasms, and hydrocephalus in the diagnosis. In differential diagnosis, structural imaging markers are now part of the criteria for other types of dementia, such as vascular, frontotemporal, dementia with Lewy bodies, and Creutzfeldt–Jakob disease [53]. Increasingly, whole-brain and hippocampal atrophy rates are being used as outcome measures in clinical trials for potential disease-modifying treatments. Future advancements may integrate imaging and cerebrospinal fluid markers, including amyloid deposition and glucose metabolism, with automated structural assessments for more accurate diagnosis and monitoring [64]. Patients presenting with behavioral and dysexecutive syndromes may have underlying AD or FTD. It was reported that patients with poor memory scores who had marked atrophy in bilateral temporoparietal regions, but limited atrophy in the frontal cortex, usually have AD, whereas those with greater frontal atrophy, less temporoparietal atrophy, and only mild memory impairment usually have FTD [59]. The identification of these subtypes, especially the extent of cortical rather than hippocampal atrophy, could be used to classify participants in a clinical treatment trial to predict the individual rates of cognitive and functional progression associated with each subtype [54,57]. Young et al. [65] have developed a machine learning method that combines the assessment of phenotypic (consisting of different subtypes) and temporal (consisting of different rates of progression) heterogeneity. They successfully used data from 819 ADNI-1 participants to identify three distinct functional progression patterns, which allowed for stratifying participants into groups, combining phenotypic and temporal heterogeneity.

### 3.2. FDG-PET

Traditional positron emission tomography (PET) detects radioactivity to quantify the level of physiological activity after the injection of a radioactive tracer. FDG PET is a biomarker for neuronal degeneration in dementia in addition to being an oncologic imaging biomarker [66]. Several studies have been published evaluating glucose use patterns in healthy subjects. Glucose use in the cerebral hemispheres is usually symmetric, and the mean glucose use patterns of the cortex, caudate, and thalamus are equal in subjects of all ages [67]. With normal aging, the largest decrease in FDG uptake has been observed bilaterally in the superior and medial frontal, motor, anterior, and middle cingulate cortices; bilateral parietal regions (with left-sided predominance); and superior and inferior parietal cortices. The metabolic rate of the superior temporal pole extending to the insular and orbitofrontal cortices is especially affected. The smallest glucose uptake decrease is observed in the bilateral medial temporal lobes (hippocampus, amygdale, and parahippocampal gyrus). The putamen, pallidum, and lateral thalamic nuclei, as well as the right posterior cingulate cortices, precuneus, bilateral occipitotemporal cortex, and cerebellum, are metabolically less impaired [68]. Clinical interpretation of FDG PET can be performed qualitatively, along with quantitative analysis to aid the reader. Any focal metabolic deficits or regional left-to-right asymmetry can identify a spatial pattern of hypometabolism compatible with specific types of dementia. Anatomic standardization of the PET image is performed by realigning the images in a stereotactic orientation using a standard brain atlas. A 3D stereotactic surface projection is then used to extract regional cortical metabolic activity, which is an alternative approach to conventional ROI analysis. The dataset is then compared with an available normal reference database using a z score formed on a pixel-by-pixel basis on the 3D stereotactic surface projection format [69]. Subjects with AD have a pattern of reduced glucose metabolism in the posterior cingulate and parietotemporal cortices in the early stages of AD, with deficits in the frontal lobes in advanced AD [70]. This pattern has a high sensitivity/specificity when comparing subjects with AD to age-matched controls. Other dementing disorders such as dementia with Lewy bodies (DLB) and frontotemporal dementia (FTD) and its variants also have a diagnostic pattern on FDG PET that is useful in the differential diagnosis of dementing disorders. Regional hypometabolism in AD, FTD, and other disorders is reflective of reduced neuronal activity and neurodegeneration [71].

### 3.3. Amyloid PET

Traditional positron emission tomography (PET) detects radioactivity to quantify the level of physiological activity after the injection of a radioactive tracer. Klunk et al. initially used the 11C-labeled Pittsburgh Compound (11C-PiB) to bind to, and visualize, amyloid plaques [72]. Concerns with the short half-life of the compound led to the development of several tracers labeled with ^18^F to bind to fibrillar amyloid aggregates, allowing the measurement of amyloid deposition in AD. There are currently three FDA-approved ^18^F radiolabeled tracers available for use in a clinical setting: [^18^F]-florbetaben, [^18^F]-florbetapir, and ^18^F-flutemetamol. All three of these tracers will show nonspecific uptake within the white matter [73]. While there are slight variations in thresholds for clinical interpretation of the amyloid PET, scans can be interpreted as positive or negative based on the ratio of white matter to gray matter radiolabel uptake. Tracer binding exclusive to white matter regions indicates a negative amyloid PET scan, while binding of the tracer in gray matter regions at equal to, or greater levels than, white matter regions indicates a positive finding (Figure 4). Amyloid PET is useful in the diagnosis of Alzheimer’s disease in the presence of cognitive impairment; however, positive imaging may be detected in individuals lacking cognitive impairment [74]. The prevalence of amyloid beta pathology in the absence of MCI increases with age and is associated with APOE genotype, with e4 carriers at higher risk for amyloid pathology. Two other methods, the SUV ratio, and the centiloid method, allow for a quantified measure of amyloid deposition. A comparison of the ratio of tracer uptake in affected cortical regions versus cortical regions that are known to be spared from amyloid deposition (such as the cerebellar cortex) provides the SUV ratio. To further quantify amyloid imaging results, the centiloid method was developed, introducing a method to analyze the intensity of cortical tracer uptake on a 0–100 scale, with 0 indicating high-certainty of amyloid-negative results, and 100 indicating results from a patient with mild to moderate AD [75].

### 3.4. Tau PET

Radiolabels have also been developed to selectively bind to tau and allow for tau PET imaging. ^18^F-T807 binding indicates the presence of tau while having low nonspecific binding to normal gray and white matter regions of the brain [76]. The deposition and accumulation of tau into neurofibrillary tangles (NFT) can be visualized using PET, allowing for in vivo recognition of the spatial distribution of tau, and the classification of patients into the Braak stages of tau pathology [77].

## 4. Blood-Based Biomarkers

The National Institute on Aging-Alzheimer’s Association Framework on AD provides biomarker profiles of individuals in the AD spectrum using three types of biomarkers—Aβ (A), tau (T), and neurodegeneration (N), known as the ATN framework. Each element of the ATN framework is rated as negative or positive and characterized as follows: A−/+, T−/+, and N−/+ [78].

The markers for “A” are determined by CSF Aβ protein 42 (Aβ42/40) and the amount of amyloid plaque in the brain through PET scanning. The “T” markers are CSF, phospho-tau (p-tau), and tau positron emission tomography. The markers for “N” are CSF total tau, atrophy on MRI measured volumetrically and by cortical thickness, and the presence of metabolic deficits on fluorodeoxyglucose (FDG) PET. Recent advancements in Alzheimer’s research have led to the development of plasma biomarkers indicating the presence of Aβ, tau, and neurodegeneration. The temporal evolution of these biomarkers was first hypothesized in 2010 and further elaborated upon by Jack et al. in 2013 [79].

As described by Jack et al. [80], the prevalence of positive biomarkers in the Mayo Clinic Study on Aging, among 1524 participants who had Aβ PET and 576 participants who had tau PET, increased exponentially with age for both Amyloid PET and Tau PET, and the prevalence exceeded the prevalence of clinically defined probable AD. Among those participants who had longitudinal evaluations, for a median of 4.8 years [81], the most rapid cognitive decline occurred in those who were A+T+(N)+, A+T+(N)−, and A+T−(N)+, as compared to participants in the other five AT(N) groups.

The prevalence of amyloid pathology, assessed using CSF measures and Aβ PET imaging among 50 to 90-year-old individuals, was found to be 2–3 times greater among APOE-ε4 allele carriers, as compared to non-carriers [74]. These investigators also found that the prevalence of Aβ positivity increased with age (from 10 to 44% among those who were cognitively normal or had subjective cognitive complaints. Among those with MCI, the prevalence increased with age from 27 to 71%, and the interval between when they first became Aβ-positive and the onset of dementia was 20 to 30 years.

Plasma biomarkers, which are both accessible and inexpensive, have recently become available and have been found to be associated with brain Aβ burden, tau pathology, and neurodegeneration, as well as the likelihood of clinical progression to dementia. In a cross-sectional and longitudinal analysis of 183 ADNI participants (CN = 97; MCI = 97), Shen et al. [82] categorized participants using cutoffs for amyloid/tau/neurodegeneration (A/T/N) plasma biomarkers, namely, amyloid-beta (Aβ) 1–42/Aβ1–40 ratio for “A”, p-tau181 for “T”, and neurofilament light for “N”. At baseline, they found participants who were A+ had a higher frequency of APOE ɛ4 carriers than those who were A−. In addition, those cognitively normal participants (*n* = 97) who were A+T+N+, or A+T+N− had faster progression compared to those who were A−T−N. In addition, among MCI participants (*n* = 86) those categorized as A+T+N+ progressed faster, with respect to cognition and the rate of brain atrophy.

More recently, Luo et al. (2022) have described the temporal evolution of ATN biomarkers among 2609 cognitively normal young to elderly participants at elevated risk for AD [83]. Among these participants, all had cognitive testing, 873 had MRI biomarkers, 519 had amyloid PET imaging and 475 had CSF biomarkers [amyloid-β42 (Aβ42), Aβ40, total tau (Tau), and phosphorylated tau181 (pTau181)] measures. Participants were subdivided by age at baseline into six age groups between 18 and 70 years and those at 70+ years. Learning effects on cognition and changes in CSF and amyloid PET biomarkers were noted in the 45–50 and 50 to 55-year age groups. Longitudinal change in the CSF Aβ42 and Aβ42/Aβ40 ratio, pTau181, and an increase in amyloid PET positivity were observed. Decreases in hippocampal volume at the baseline age of 55–60 years and a decline in cortical thickness and cognition were found at the baseline age of 65–70 years. The rate of change in the Aβ42/Aβ40 ratio, Tau, pTau181, and amyloid PET positivity appeared to peak in the 65–70 age group, although the rate of decline of hippocampal volume continued to accelerate. Among APOE ɛ4 carriers, the rates of change for all CSF biomarkers, amyloid PET measures, and cognition were steeper.

The inter-relatedness between neuropathological and plasma biomarkers was described in a study by Grothe et al., 2021 [84]. In this study, CSF biomarkers obtained ante mortem in 45 individuals were related to “standardized postmortem assessments of AD and non-AD neuropathologic changes at autopsy” and to reference biomarker values obtained from amyloid-PET-negative healthy controls. It was found that “CSF biomarkers, in vivo, detected neuropathological changes with high discriminative value”. Optimal biomarker cutoffs were derived for Aβ1–42, t-tau, and p-tau181. In a subset of individuals, it was found that plasma p-tau181 also appeared to discriminate neuropathological features. These findings add confidence to the use of CSF and plasma biomarkers in providing accurate and scalable values, representing the presence and severity of neuropathological features of AD.

The methodologies used to measure the plasma Aβ42/40 ratio have been found to have very variable discriminative accuracies, as was identified in a head-to-head comparison of eight different plasma Aβ assays compared to findings from Aβ PET imaging and CSF Aβ42/40 from two different cohorts (Biofinder and ADNI) [85]. The results from this study indicated that mass-spectrometry-based studies performed significantly better than most immunoassays for plasma Aβ42/40 amyloid-PET-positivity and stronger correlations with tau-PET signal. Phosphotau plasma biomarkers (p-tau181 and especially, p-tau217, are biomarkers of both amyloid and tau pathology, even in an early preclinical stage of AD [86].

Structural and functional MRI, FDG-PET, Aβ and tau PET imaging, in combination with demographic, genetic, and clinical data, from the ADNI longitudinal database, studied by Veitch et al. [87] identified several important findings among participants who were cognitively normal but were amyloid positive: (a) Aβ deposition occurs concurrently with functional connectivity changes, including disconnection within the default mode network; (b) functional connectivity, volumetric measures, regional hypometabolism, and even cognitive changes were detectable at subthreshold levels of Aβ deposition; (c) a specific temporal and spatial pattern of tau pathology, on tau PET, is related to subsequent cognitive decline; (d) vascular pathology is related to AD progression by both Aβ-dependent and independent mechanisms; and (e) cerebrovascular disease and the APOE ε4 allele interact to impede Aβ clearance.

### 4.1. Amyloid Beta and Ratios

Numerous studies have evaluated the plasma levels of Aβ42, Aβ40, and the Aβ42/40 ratio in patients with AD using different testing platforms. Most of these studies have reported lower plasma Aβ levels in AD patients compared to cognitively unimpaired individuals. Additionally, it has been observed that the accuracy of distinguishing abnormal from normal amyloid-β positron emission tomography (Aβ-PET) scans increases gradually [88,89].

Leuzy et al. demonstrated that using Aβ-PET status to identify AD pathology results in more accurate plasma Aβ quantification compared to clinical diagnosis [90]. Janedlize et al. demonstrated that quantifying plasma Aβ using automated immunoassays and MS-based methods accurately predicts brain Aβ burden using the CSF Aβ42/40 ratio or Aβ-PET as gold standards [91]. The findings remain consistent despite the use of different analytical methods. Although there are variations in detection sensitivity and biomarker concentrations, MS-based techniques tend to perform better than immunoassays in most studies. In a comparative study involving ten different assays, LC-MS had the most effective diagnostic performance. The plasma Aβ42/40 ratio shows a stronger correlation with brain Aβ burden and provides better accuracy for diagnosis and prediction than either Aβ42 or Aβ40 alone [86].

In a study conducted by Shoji et al., it was found that patients with Alzheimer’s disease (AD) had significantly higher levels of tau protein in their cerebrospinal fluid (CSF) compared to the control group of normal individuals. However, the levels of Aβ40 protein did not show any significant difference between the two groups. Interestingly, the Aβ42/40 ratio was identified as a more reliable indicator for AD detection [92].

Lewczuk et al. analyzed the concentrations of Aβ42, Aβ40, and total tau (T-tau). While the levels of Aβ40 did not vary significantly among the groups, the Aβ42/40 ratio provided more accurate classifications than Aβ42 alone but without statistical significance [93]. Nutu et al. found that the Aβ42/40 ratio was highly effective in differentiating AD from Parkinson’s disease dementia (PDD) and DLB [94].

### 4.2. Neurofilament Light (NfL)

Plasma neurofilament light (pNfL) is elevated with age and with multiple neurological conditions, including AD and frontotemporal dementia (FTD) in both CSF and plasma [95,96]. In an ethnically diverse cohort, Barker et al. studied 309 older participants initially categorized as cognitively normal (CN) or cognitively impaired [96]. The research primarily targeted Alzheimer’s disease (AD) while including other neurological and neuropsychiatric disorders. The study observed a positive association between pNfL levels and age, degree of hippocampal atrophy, amyloid burden, and both cognitive and functional status. Female CN subjects showed higher pNfL levels. pNfL levels were not elevated in neuropsychiatric disorders accompanied by cognitive impairment. The study also showed that higher baseline levels of pNfL indicated future cognitive and functional decline above that predicted by hippocampal atrophy and baseline memory scores. While pNfL may not be a stand-alone diagnostic tool, it has potential as a supplementary measure to neuroimaging and cognitive tests, especially when extensive neuroimaging is unavailable [97]. The biomarker shows promise in differentiating neurodegenerative diseases from neuropsychiatric disorders [96].

### 4.3. Plasma Tau Protein

Although elevated brain Aβ levels can be observed in many individuals without cognitive decline, tau protein accumulation is a more reliable indicator of cognitive deterioration and is strongly associated with Alzheimer’s disease (AD) diagnosis. However, total tau levels can be ambiguous, as they are not specific to various neuropsychiatric conditions, including traumatic brain injury. Instead, current assays focus on specific phosphorylated tau proteins, namely p-tau181, p-tau217, and p-tau231, which are far more specific to AD pathology [98].

Recent advances in sensitive assays and mass spectrometry technologies have significantly improved the identification of these p-tau species in plasma, correlating strongly with AD’s neuropathological changes. Research shows that plasma p-tau217 levels rise earlier in AD’s pathological spectrum, showing higher diagnostic accuracy than p-tau181 in identifying individuals at risk for AD. Generally, plasma p-tau levels are markedly higher in AD patients compared to those who are cognitively normal [98,99]. Meta-analyses and studies in population-based cohorts have supported the efficacy of plasma p-tau biomarkers, particularly p-tau217, in predicting the transition from mild cognitive impairment to dementia, even when considering other risk factors [100,101]. Importantly, even minor elevations in plasma p-tau levels correlate with future cognitive decline and are indicative of brain atrophy and reduced glucose metabolism, independent of elevated brain Aβ levels [102].

In a recent study, it was observed that APOEε4 potentiated Aβ effects on tau accumulation [103]. This process occurred through pathological tau phosphorylation, assessed with plasma p-tau-217. They found that plasma p-tau and tau-PET were not interchangeable biomarkers for detecting tau pathology. Rather, the appearance of p-tau biomarkers, representing hyperphosphorylated tau, occurs in tandem with the development of Aβ pathology and tau-PET becomes positive once aggregated tau forms neurofibrillary tangles. As such, fluid biomarker changes precede imaging tau biomarkers because plasma p-tau biomarkers measure soluble and non-aggregated hyperphosphorylated tau fragments, whereas tau-PET detects insoluble tau deposits. Further, ApoE4 expression appears to increase tau phosphorylation [103].

### 4.4. GFAP

The activation of astroglia and astrocytosis has been identified as a potential bridge that connects amyloid and tau pathologies in Alzheimer’s disease (AD) [104]. Early research by Oeckl et al. discovered that AD patients had elevated levels of plasma glial fibrillary acidic protein (GFAP), which were notably correlated with cognitive decline [105]. Subsequent studies have emphasized GFAP’s significance as an essential biomarker in AD research, although the data show substantial variability. For instance, Pereira et al. demonstrated that GFAP levels increased in subjects with amyloid pathology and were capable of predicting amyloid beta positron emission tomography (Aβ-PET) positivity with an area under the curve (AUC) of 0.76, outperforming cerebrospinal fluid (CSF) GFAP and other glial markers [104]. Interestingly, the correlation between plasma GFAP levels and Aβ pathology appeared stronger than that between CSF GFAP levels and the same pathology. Moreover, elevated plasma GFAP levels have been linked to brain amyloid pathology, regardless of cognitive status, whether cognitively normal, experiencing mild cognitive impairment (MCI), or clinically diagnosed with dementia [104].

Combining plasma GFAP levels with other AD-associated biomarkers can significantly enhance the diagnostic precision of individual tests. One study reported an AUC of 0.78 when evaluating amyloid pathology solely based on GFAP levels. However, when plasma GFAP was analyzed in conjunction with the plasma Aβ42/40 ratio, the AUC increased to a remarkable 0.92 [106]. Additionally, a biomarker panel comprising plasma GFAP and neurofilament light chain (NfL), along with other established AD risk factors like age, sex, and APOE genotype, yielded an AUC of 0.91 in differentiating cognitively unimpaired from AD dementia subjects; an AUC of 0.81 for differentiating cognitively unimpaired from MCI subjects; and an AUC of 0.87 in predicting Aβ positivity [107].

### 4.5. Beta-Synuclein

Advancements in assays have enhanced the sensitivity and specificity for measuring blood levels of β-synuclein, a synaptic biomarker. Elevated levels of this biomarker serve as early indicators of synaptic degeneration in Alzheimer’s disease (AD) and are associated with plasma levels of p-tau181 and Aβ, suggesting a link to amyloid pathology. Importantly, β-synuclein levels were not altered in other tauopathies, and its associated brain structural changes differ from those linked to plasma NfL and p-tau181. These insights position β-synuclein as a critical tool for the early diagnosis of AD [108,109,110,111,112,113,114].

### 4.6. APOE ɛ4 Allele

The presence of the ɛ4 allele of the apolipoprotein E (APOE) gene is a strong genetic factor that predisposes individuals to sporadic Alzheimer’s disease (AD) [115]. However, relying solely on APOE status for diagnosing AD is not sufficient [116,117]. The PrecivityAD™ test combines plasma Aβ42/40 ratio, APOE status, and age to predict the likelihood of amyloid plaque presence and is an example of how combining different biomarkers is more effective than using only one [118].

### 4.7. Platelet-Derived Amyloid-β Protein Precursor (AβPP)

Platelets have emerged as an important source of AD biomarkers [119]. They contain almost all circulating amyloid-β protein precursor (AβPP), and the ratio of two specific forms of AβPP (130/110 kDa) shows reliability as an indicator of AD diagnosis, independent of age, and is correlated with cognitive decline [120]. Furthermore, platelet-derived tau levels have been linked to AD-related brain atrophy and clinical outcomes as measured by the Clinical Dementia Rating (CDR) scale [121,122].

In a groundbreaking study, Guzmán-Martínez et al. introduced the Alz-tau^®^ biomarker, which is based on the ratio of heavy tau (HMWtau) to low-molecular-weight tau (LMWtau) in human platelets [123]. This ratio correlates with reduced brain volume, as determined through structural MRI, and provides a new, promising avenue in AD diagnostics. However, the efficacy of the Alz-tau^®^ marker needs further validation, as it has not yet been directly compared with established AD biomarkers such as Aβ-PET and cerebrospinal fluid (CSF) measures [124]. More studies are needed to investigate its potential as a diagnostic tool.

### 4.8. Future of Plasma-Based Biomarkers in Alzheimer’s Research

Plasma biomarkers, such as Aβ 42/40 ratio and phosphorylated -tau (p-tau) levels have recently had a significant impact on research in the field of Alzheimer’s disease. While novel biomarkers and PET tracers are still under investigation, the first identification of plasma biomarkers was undoubtedly a game-changer, spearheading the field toward real-world applications. These biomarkers are more effective than clinical evaluations in predicting outcomes for individuals with subjective cognitive decline (SCD) and mild cognitive impairment (MCI) [125]. However, the field is still grappling with determining which plasma biomarker or combination of biomarkers is most effective in identifying AD pathology. Recent research shows that using Simoa, MSD, or MS-based methods gives the best results for measuring plasma p-tau231 and p-tau217. These markers are more closely linked to early signs of Aβ disease, and its long-term build-up compared to other plasma biomarkers [126].

Studies have shown that combining various biomarkers into a single panel could improve diagnostic accuracy. For example, a model integrating plasma Aβ42/40, plasma p-tau181, and APOE status demonstrated impressive prediction capabilities for Aβ positivity and AD progression over six years. Nonetheless, the cost is still a barrier to widespread clinical adoption as the techniques used, Elecsys-based fully automated assays, are costly. Even though using multiple biomarkers would be expensive and could limit widespread use, combining different plasma biomarkers might be the most effective way to identify patients in the early stages of Alzheimer’s disease [127].

The fast-paced developments in this domain focus on creating a clinically accessible tool for timely AD diagnosis, especially during its preclinical stages. However, several challenges still hamper progress, such as the absence of universally accepted reference values, significant inter-method variability, and the high cost associated with the most promising methods. To address these issues, future studies must perform head-to-head comparisons between different assays and analytical methods. They should also adhere to standardized operating procedures to minimize biases and include diverse, underrepresented populations for more comprehensive data [128].

Moreover, combining biomarkers with other risk factors enhances predictive accuracy. Studies utilizing a panel of biomarkers based on established AD indicators have shown promising results, although cost-effective automated assays are essential for broader implementation [129]. Investment in fully automated assays has mitigated the high cost associated with the most promising methods, and many groups have developed accurate in-house reference values. However, a consensus is urgently needed to facilitate meaningful advancements in the field. The EU/US CTAD Task Force meeting in May 2022 marked a step toward this goal, highlighting scientific progress, current limitations, and future directions for implementing these biomarkers in clinical settings [130]. In conclusion, the use of plasma biomarkers for AD diagnosis is rapidly evolving, driven by advancements in analytical methods and a focus on real-world applicability. These developments pave the way for a future where plasma biomarkers can be widely used in AD diagnostics. However, further investigation is crucial for universal acceptance and implementation.

### 4.9. Ethnic Studies Regarding Plasma-Based Biomarkers and AD

Asken et al. conducted a study to evaluate the associations between common plasma biomarkers for Alzheimer’s disease in a diverse cohort of older adults. The study included 379 participants with an average age of 71.9 years, of which 60.2% were female, and 57% were of Hispanic ethnicity, predominantly of Cuban or South American ancestry. The biomarkers evaluated in the study were p—tau181, GFAP, and NfL. Additionally, 240 participants completed Aβ-PET scans [131]. The results showed that p-tau181 levels were significantly higher in individuals with amnestic mild cognitive impairment (MCI) and dementia than those who were clinically normal. Furthermore, p-tau181 demonstrated superior capabilities in distinguishing between positive and negative Aβ-PET outcomes, outperforming other biomarkers like GFAP and NfL. Interestingly, the data revealed no significant interactions between biomarker performance and ethnicity. However, within the amnestic MCI group, Hispanic participants showed lower odds of elevated p-tau181 levels than non-Hispanics, suggesting the possibility of non-Alzheimer’s factors contributing to memory loss in this ethnic group. The study indicates that plasma p-tau181 could be a valuable marker for cognitive impairment across ethnicities. It also suggests that Hispanics may have a higher likelihood of non-Alzheimer-related memory loss.

Schindler et al. conducted a study to assess plasma biomarkers’ effectiveness in detecting brain amyloidosis across different racial groups, explicitly comparing African American (AA) and non-Hispanic White (NHW) individuals [132]. A total of 152 participants were matched by age, APOE ε4 carrier status, and cognitive function, undergoing blood and CSF collection and amyloid PET scans in a subset. The results showed that AAs were less likely than NHWs to have brain amyloidosis, based on CSF Aβ42/Aβ40 levels. However, the study found that race did not significantly affect the diagnostic accuracy of plasma Aβ42/Aβ40 biomarkers in predicting CSF Aβ42/Aβ40 status. In contrast, plasma p-tau181, p-tau231, and NfL biomarkers were less reliable in predicting amyloidosis in AAs, leading to a potential risk of misdiagnosis. Therefore, while plasma Aβ42/Aβ40 assays may offer a consistent measure for detecting brain amyloidosis across racial groups, caution is needed when using p-tau and NfL biomarkers due to their potential for inconsistent performance.

In conclusion, this review article summarizes current knowledge regarding the evolution of neuropathological changes in the brain and how it relates to commonly used and available brain imaging and fluid biomarker changes in the CSF and plasma. The development of plasma biomarkers, in particular, provides the potential for convenient, non-invasive markers of the underlying pathology in AD. The continuing development of, and wider access to, these plasma biomarkers will enable more effective ways to routinely screen for AD pathology, as well as improve clinical diagnosis of AD, in very early and even preclinical stages of the disease. The inclusion of these plasma biomarkers in clinical trials has shown their potential for determining candidacy for clinical trials, tracking the progression of the disease, and may allow for a convenient way to assess the effectiveness of various interventions for modifying the underlying pathology of AD.

## Figures and Tables

**Figure 1 diagnostics-14-00704-f001:**
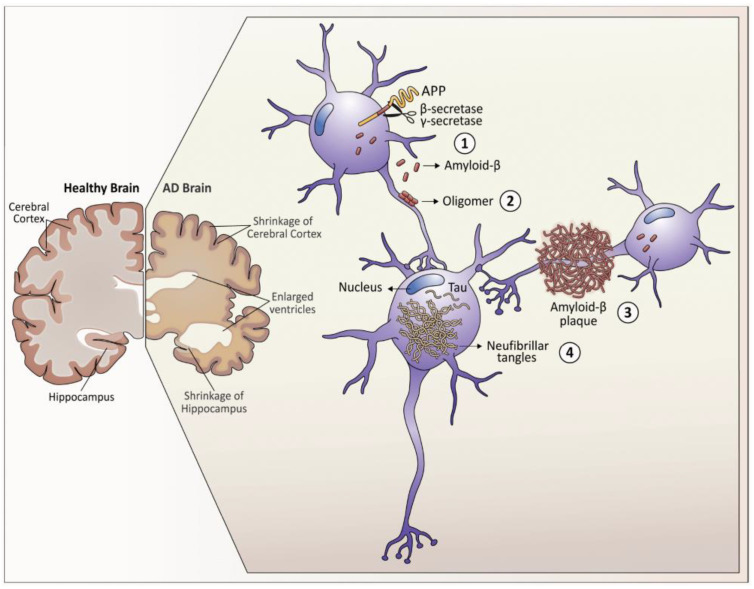
Neuropathological hallmarks that characterize Alzheimer’s disease. As Alzheimer’s disease progresses, the brain tissue shrinks, and the volume of the ventricles, which contains cerebrospinal fluid, increases markedly. At the molecular level: 1. Amyloid-β peptides are produced by the cleavage of the amyloid precursor protein (APP) in the membrane of the neurons. 2. In the space between the neurons, amyloid-β forms oligomers that are thought to disrupt the function of the synapses and receptors present in the neuronal plasma membrane. 3. The fibrils of the amyloid-β oligomers in plaques interfere with the function of the neurons. 4. Tau hyperphosphorylation causes neurofibrillary tangles within neurons, displacing intracellular organelles and disrupting vesicular transport. Reproduced with permission from [25], published by Impact Journals, LLC, 2020.

**Figure 2 diagnostics-14-00704-f002:**
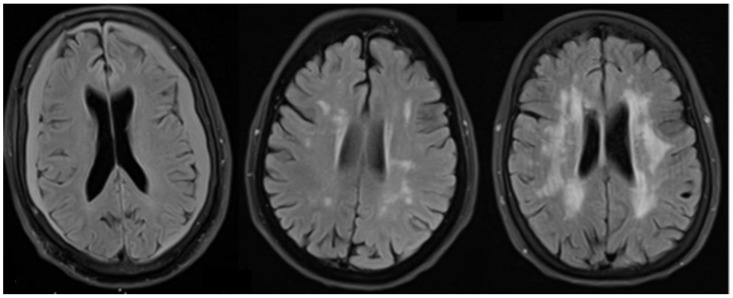
T2-FLAIR brain MRI scans show from left to right: no white matter hyperintensities (WMHs) in an elderly subject with no cognitive impairment; mild to moderate WMHs in a subject with mild cognitive impairment; and moderate to severe WMHs in a subject with mild Alzheimer’s disease.

**Figure 3 diagnostics-14-00704-f003:**
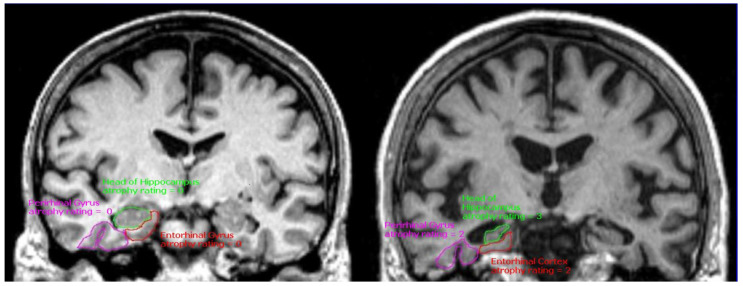
T1-weighted MPRAGE MRI images of a cognitively normal 65-year-old (**left**) and a 75-year-old (**right**) with mild to moderate Alzheimer’s disease. The image on the left shows no atrophy in the medial temporal regions, while the image on the right shows moderate atrophy in both the entorhinal cortex (red) and perirhinal gyrus (purple), and severe atrophy in the hippocampus (green). This image also shows generalized cortical atrophy, especially in the insular region, along with enlargement of the lateral and third ventricles.

**Figure 4 diagnostics-14-00704-f004:**
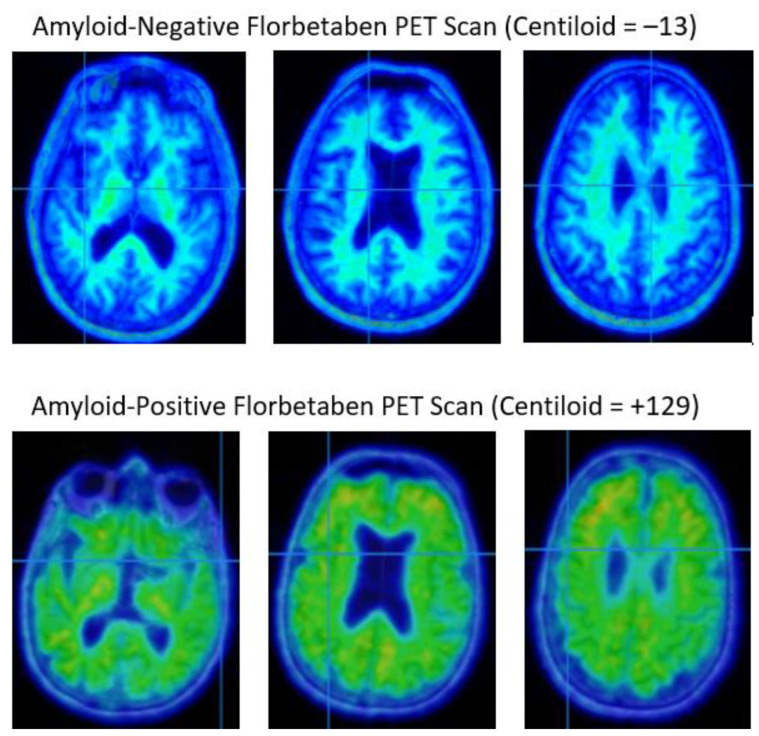
[^18^F]florabetaben PET scans. The top row shows a negative PET scan characterized by the binding of the tracer to the white matter only, with an absence of the tracer binding in the cortical gray matter and a clear contrast between white and gray matter. The bottom row shows a positive PET scan in a subject with Alzheimer’s disease, characterized by binding of the tracer in both white matter and cortical gray matter, with loss of gray–white matter contrast.

## Data Availability

Not applicable.

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
