# Peer review of "Neuropathology, Neuroimaging, and Fluid Biomarkers in Alzheimer’s Disease"

_diagnostics, 2024, doi:10.3390/diagnostics14070704_

Round 1

Reviewer 1 Report (Previous Reviewer 1)

Comments and Suggestions for Authors

I am happy with the changes that have been made.

Author Response

No additional revisions were requested by this reviewer.

Reviewer 2 Report (Previous Reviewer 4)

Comments and Suggestions for Authors

A very interesting and modern topic. It could definitely be of relevance for the readers of the journal.

I think it could add some nice value to the diagnostics journal.

Some minor suggestions, though:

I think it will be nice if authors will give some extra explnations in the Introduction on why they focused on Alzheimer’s disease (AD), Parkinson’s 21 disease (PD), and amyotrophic lateral sclerosis (ALS).

It seems obvious for a specialist, but I think is the right think to do in the Introduction.

Also I would limit the conclusions to a few phrases and separate the future perspective subchapter.

the authors did a wonderful work discussing these matters above, so maybe these lats 2 chapters should be separated.

Comments on the Quality of English Language

English is fine

Author Response

There were no references in the introduction to Parkinson’s disease or amyotrophic lateral sclerosis (ALS).

There was no "future perspective subchapter" in our paper.

This manuscript is a resubmission of an earlier submission. The following is a list of the peer review reports and author responses from that submission.

Round 1

Reviewer 1 Report

Comments and Suggestions for Authors

This review addresses an important and interesting set of issues related to the neuropathology and diagnosis of Alzheimer’s disease.

In the abstract, the authors indicate their intention to ‘review how the AD diagnostic process has been transformed…’

The manuscript covers a lot of ground and provides for the most part a useful primer/update on the pathology and biomarkers, however I have some suggestions for possible improvements.

The reference section/citations are a mess in the sense of being out of sequence (e.g. Section 1 ends with reference 17 and Section 2 starts with reference 28 with the next reference cited as 148 which is odd because the last reference in the reference list is 136), some references are incomplete in the reference list (21, 41, there may be others).  Sentences end in commas (line 79) or lack a full stop (line 116).

Section 2.7  

Some portion of a sentence may be missing, (ending ‘interstitial solutesMultiple’). References 39,40 are not cited.

2.8

Citation 21 is incomplete in the reference list. Not clear that reference listed as 39 is ever cited. The quotations marks are incomplete line 255. Also, the source of the quotations is unclear. Why quote, rather than paraphrase the conclusions?

3.1 Structural MRI.

There are no separate paragraphs in this long section. They could be used to help impose better order. It seems to jump around on themes of imaging pathology correlations, imaging clinical correlations, imaging and prognostication (in AD), imaging and differential diagnosis, circling back often to a previous theme.

Multiple assertions without supporting citations lines 280-290. Does reference 57 serve to support these? If so, suggest cite this earlier, if not provide citations, even if review article.

Passing reference is made to new MRI techniques: e.g. resonance spectroscopy, DWI , arterial spin labelling. It is not in the spirit of bringing the reader up to date to have the only reference relevant to those comments a paper from 2005. There are much more recent review papers that could have been cited.

The diagnostic utility of imaging biomarkers vs that of fluid biomarkers in relation to the phase of evolution of AD seems to have been addressed with the single sentence ‘ However, in the transition from asymptomatic to MCI stages markers of amyloid pathology often show more pronounced abnormalities than structural markers’. While the Clifford Jack biomarker figure, shown at every AD conference for many years, is certainly not obligatory (!), the sequencing of biomarker sensitivity – where they start to become informative-- warrants more attention, in my view, than this cursory treatment.

The claim that volumetric measures from MRI scans can be used to classify neurodegenerative subtypes is supported by an article in which amyloid positivity had been established (i.e. preliminary support for AD existed). I suggest reconsider the statement or provide additional context for it. In general, it is sometimes unclear who the review is targeting: the clinician who might be seeking guidance on the diagnostic/prognostic value of a single scan on an as yet undiagnosed patient or researchers with cohorts of patients with established diagnoses of AD.

3.2

No mention of the utility of FDG PET for differentiating different dementing syndromes/conditions is made in this section. This seems like a missed opportunity. There are several recent reviews that cover this that could be cited. The citations for this section, 3.2. , (citations 72-75) range in date from 1982 to 2009 and hardly constitute an updating for an interested reader.

3.3.

No mention of amyloid scan results in non-AD dementing syndromes (e.g. Lewy body disorders)

Why no mention of CSF biomarkers, other than under the heading of ‘Blood-based Biomarkers?

Section 4

Was line 446 Plasma Based AD and Neurodegenerative Biomarker Profiles supposed to be a heading?

Line 470 seems to have names out of place. Line 472 suggest you state that phosphotau217 (p- tau217) is a plasma assay.

Line 494  ‘Additionally, it has been observed that the accuracy of distinguishing abnormal  from normal amyloid-β positron emission tomography (Aβ-PET) scans increases gradually’ . What does this mean? Suggest rephrase. Similarly for the next sentence ‘Leuzy et al. demonstrated that using Aβ-PET status to identify AD pathology results in more accurate plasma Aβ quantification compared to clinical diagnosis’ : the meaning of this sentence is unclear.

Comments on the Quality of English Language

The English is satisfactory, however, there is a need for greater clarity on a number of occasions as I explicitly state. 

Reviewer 2 Report

Comments and Suggestions for Authors

Alzheimer’s disease (AD) is a severe neurodegenerative disease affecting thousands of people worldwide.  Although there is no specific therapy to cure AD, early diagnosis and intervention of AD are useful in delaying AD progress. In this paper, the authors reviewed how the AD diagnostic process has progressed in recent years through the use of neuroimaging and fluid biomarkers. This paper helps clinicians enhance their understanding of AD pathophysiologic mechanisms and early diagnosis. However, there is a lack of intrinsic connection between the various parts of the article. This article is only a simple list of neuropathological changes and biomarkers of AD progress in recent years and lacks a systematic summary. In general, this paper has several major problems as follows.

1. In the subsection 2, the authors listed several neuropathological changes of AD. However, the authors did not explain or discuss the linkages or casual associations between individual neuropathological changes.

2. Authors can add a figure to further explain the relationship among all mentioned pathophysiologic changes of AD.

3. What is the underlying linkage between subsection 2 and subsection 3? Do the neuroimaging and fluid biomarkers in aging and dementia correspond to these pathophysiologic changes of AD? What are the likely future trends in biomarkers?

4. Are copyrights acquired when reprinting the figures in this paper? This should be stated under each figure.

Comments on the Quality of English Language

This paper uses correct English terminology and flows well. It is recommended that the authors strengthen the strings between the parts of the paper.

Reviewer 3 Report

Comments and Suggestions for Authors

Thank you for giving me the opportunity to review the manuscript entitled "Neuropathologiy, Neuroimaging and Fluid Biomarkers in Alzheimer's Disease" in which the authors presents a comprehensive review of the three topics addressed in the title in patients with AD.

The manuscript is well written and structured and addresses an important topic. 

Although, there are some issues that need to be addressed:

- The few images are of good quality but they are all represents from previous publications. However, the images are not mentioned in the text.

- In the imaging section, the authors address structural MRI and volumetric MRI. An image dealing with progressive hippocampal atrophy in patients with MCI progressing to AD would be beneficial. The authors should also address the importance of (semi-)automatic segmentation and volumetric analyses of hippocampal structures, Furthermore, the role of ASL should be more emphasized as well as functional MRI.

- the citation of the literature is not consistent throughout the manuscript and should be adjusted according to the guidelines of the manuscript.

Reviewer 4 Report

Comments and Suggestions for Authors

A good theoretical report on the Neuropathology, Neuroimaging and Fluid Biomarkers in AD.

Well written piece.

Definetlly it is of inters for the readers.

minor suggestions:

- the general part on AD, including the pics of NGT and amyloid plaques can be reduced in size.

Please finish with some concluding remarks, rather then with the Ethnic studies.
